# Galvanic Skin Response Features in Psychiatry and Mental Disorders: A Narrative Review

**DOI:** 10.3390/ijerph192013428

**Published:** 2022-10-18

**Authors:** Renata Markiewicz, Agnieszka Markiewicz-Gospodarek, Beata Dobrowolska

**Affiliations:** 1Department of Neurology, Neurological and Psychiatric Nursing, Medical University of Lublin, 20-093 Lublin, Poland; 2Department of Human Anatomy, Medical University of Lublin, 20-090 Lublin, Poland; 3Department of Holistic Care and Management in Nursing, Medical University of Lublin, 20-081 Lublin, Poland

**Keywords:** galvanic skin response (GSR), electrodermal activity (EDA), biofeedback, mental disorders

## Abstract

This narrative review is aimed at presenting the galvanic skin response (GSR) Biofeedback method and possibilities for its application in persons with mental disorders as a modern form of neurorehabilitation. In the treatment of mental disorders of various backgrounds and courses, attention is focused on methods that would combine pharmacological treatment with therapies improving functioning. Currently, the focus is on neuronal mechanisms which, being physiological markers, offer opportunities for correction of existing deficits. One such indicator is electrodermal activity (EDA), providing information about emotions, cognitive processes, and behavior, and thus, about the function of various brain regions. Measurement of the galvanic skin response (GSR), both skin conductance level (SCL) and skin conductance responses (SCR), is used in diagnostics and treatment of mental disorders, and the training method itself, based on GSR Biofeedback, allows for modulation of the emotional state depending on needs occurring. Summary: It is relatively probable that neurorehabilitation based on GSR-BF is a method worth noticing, which—in the future—can represent an interesting area of rehabilitation supplementing a comprehensive treatment for people with mental disorders.

## 1. Introduction

It has been proven that skin conductance, as a physiological indicator of mental processes, is an important diagnostic parameter in the determination of mental disorders. This method was initiated by Fere, who described the galvanic skin response (GSR) in 1888. He found that external or internal stimuli result in the development of potential which, through the parasympathetic autonomic nervous system, determines an emotional status expressed as skin resistance [1]. His observations were confirmed in further studies by Jung, who considered GSR to be a “psychogalvanic reflex” [2].

To understand a relationship between mental activity and an electrodermal change, mechanisms resulting from physiological body reactions should be understood. Skin resistance and changes in conductance are associated with physiological processes, being proof of the joint functioning of different brain structures. The most important of these structures are the premotor cortex, the hypothalamus, the limbic system (the hippocampus, the amygdala, the anterior nuclei of thalamus, and mammillary body), and the reticular formation [3].

From the mental disorders point-of-view, an important role is played by the limbic system, as it is responsible for emotional processes, learning processes, and motivation. It is formed by a network of interconnected neurones which, by communicating with the frontal, temporal, and parietal lobes, enables reception of emotions. The thalamus is another important connection with the limbic system, as it controls the autonomic nervous system (sympathetic and parasympathetic) and the endocrine system [3].

Cooperation of these two areas is important because it guarantees internal homoeostasis and correct functioning of the Amygdala–Hypothalamus–Pituitary–Adrenal (AHPA) axis, responsible for neurophysiological stress reactions. Any disruptions in its function (hyperfunction, hypofunction) result in reduced cognitive function, insomnia, rumination, fear, and depression [3,4,5].

In patients with mental disorders, these deficits occur with varied intensity, depending on the clinical diagnosis. They concern cognitive (working and verbal memory, attention, information processing, learning), executive, language, as well as visual and spatial functions. All have a negative impact on patients’ quality of life [6,7,8,9].

It should be emphasised that antipsychotics mainly eliminate productive symptoms, but they do not improve cognitive functions, and restoring them is an important element in comprehensive treatment [10,11,12,13,14].

There are various rehabilitation methods available. One of them is GSR Biofeedback (GSR-BF), whose positive effects may effectively eliminate cognitive deficits in patients with mental disorders, and with the developed training profile and management standard, may facilitate inclusion of this method in comprehensive rehabilitation treatment [15].

The aim of this paper is to discuss electrodermal activity (EDA), the use of the galvanic skin response (GSR) in psychiatry (skin conductance responses (SCR) as a symptoms indicator in mental disorders), and a relationship resulting from the method applied, being, at the same time, a neurophysiological effect of changes occurring during the therapy.

Critical review of available literature was performed. Three databases were searched—PubMed, Scopus, and Web of Science—for publications with research results on use of galvanic skin response (GSR) in diagnostics and rehabilitation of mental disorders. When searching, combinations of the following keywords were used: galvanic skin response, electrodermal activity, biofeedback, and mental disorders.

## 2. Electrodermal Activity (EDA)

Fere described the galvanic skin response (GSR) [1] and found that external (sound) or internal (cognitive processes) stimuli result in development of potential which, through the autonomic nervous system (ANS), expresses a relevant emotional status registered in a graphic recording of skin resistance (electrodermal reaction) [16,17].

The plot of changes depends on the activity of sweat glands which are markers providing information on the psychophysical condition of the body. In a situation of stress, tension, or anxiety, sweat secretion increases, and it becomes a conductor, reducing the skin resistance. The situation is the opposite in cases of satisfaction or relaxation; then, conductance is reduced, and skin resistance increases [18].

The galvanic skin response (GSR) registers physiological electrodermal activity, and it involves two measurements, depending on the source: endosomatic and exosomatic. The endosomatic involves one measurement, so-called skin potential (SP), considered to be a skin conductance response. The exosomatic measurement creates a closed circuit consisting of a galvanometer, an electric battery, and a human body, and uses either direct (DC) or alternating (AC) current. The exosomatic method with direct current (DC) is the most used method for measurements of electrodermal activity (EDA) [19,20,21].

Two main components are distinguished in EDA evaluation: a skin conductance level (SCL) and the previously mentioned skin conductance responses (SCR). These two components are generated in different areas of the brain, depending on source of origin, stimulus type, and emotional aspect [20].

The skin conductance response is determined by an electrode response, associated with events, specific stimuli, or sources, accompanied by emotions. Its characteristic feature is a short recording, both in terms of its amplitude (response magnitude) and frequency (number of responses). The skin conductance level has a longer recording, of horizontal course, and is expressed mainly by its amplitude [22,23,24].

In evaluation of electrodermal activity (EDA), important aspects include initial values, general SCL values (the skin conductance level), and the skin conductance responses, SCRs (change size) [1,16]. They are expressed as kiloohms or microsiemens (microsiemens are a reciprocal of resistance; 1000 kOhm = 1 microsiemens).

## 3. Use of Galvanic Skin Response (GSR) in Psychiatry

The galvanic skin response (GSR) is used mainly in psychological and medical tests. In psychiatry, it is used to evaluate anxiety disorders [15,25,26], depression disorders [27,28,29,30], suicidal tendencies [31,32], bipolar affective disorder [33,34,35], and schizophrenia [36,37,38]. Table 1 presents mental disorders and the EDA profile (SCL and SCRs) in selected clinical units. 

An analysis of the GSR profile is based on data of initial values, phase SCRs, and the skin conductance level. These very sensitive indicators reveal many common features in specific clinical conditions, which are important for further treatment from the diagnostic point-of-view. On the basis of SCL and SCRs, it is possible to determine the disease stage, symptom intensity, and possibility of recurrence. The research shows that in affective disorders, electrodermal activity differs mainly in its SCL recording; it is reduced in depression, and increased in mania [15,28,31,37,39]. This is confirmed by research in which a group of women with bipolar disorders in remission was compared with a group of healthy women using presentation of words with positive and negative emotional load. The author’s studies indicate that both the healthy group and the group diagnosed with bipolar disorders in remission do not show significant differences in their skin conductance levels, SCL. A difference in both groups concerned mainly the initial SCL values, which are lower in the group of women with bipolar disorders [40].

Similar results were obtained when electrodermal activity was measured using sounds as external non-emotional stimuli to evoke emotional responses (fear) in patients with depressive disorders. The research of this author indicates that the emotional condition which developed in response to the sound resulted in stronger skin conductance responses, SCRs, in a group of persons with depression versus the control group [41]. Possibly, fear inspired numerous conductance responses associated with emotional reactions [42]. Studies have been performed on electrodermal activity in patients with depressive disorders and in healthy people by presenting them with images of faces of happy and neutral people. The research indicates that patients with depressive disorders are characterised by reduced skin conductance responses (SCRs) versus persons in the control group [43].

Interesting research results were obtained by researchers who compared groups of patients with clinically diagnosed schizophrenia and depression in terms of the initial values and general skin conductance levels, SCL, on the basis of mathematical tests. The research indicated that schizophrenia patients had higher SCL values in terms of responses to skin conductance when compared to patients with depression—whose SCL profile is flat—and initial results are low [44,45,46]. The flat recording in the group of patients with depression may possibly result from lack of activation and problems with relaxation (physiological flexibility deficit).

Studies on a group of schizophrenia patients confirm these results. They indicate that in schizophrenia patients, electrodermal activity, EDA, differs in terms of skin conductance responses, SCRs. In patients with positive symptoms, numerous skin conductance responses, SCRs (both spontaneous and non-specific, NS.SCR), are visible versus people in whom negative symptoms predominate, where their reduction is visible. In addition, in terms of general SCL activity and initial values, the graphic recording differs. Patients diagnosed with schizophrenia in remission differ significantly in these two indicators. In these persons, the initial value is usually lower or similar to mean relative changes (this also concerns values associated with activation) [23,47,48]. Below there are photos from our own research showing changes in the SCL and SCRs values (EDA profile) in a patient with diagnosed schizophrenia (Figure 1 and Figure 2).

Therefore, when scant skin conductance responses, SCRs, transform into numerous skin conductance responses, and lower activation values raise significantly, the remission is then possibly progressing to the disease-exacerbation phase (condition of increased reactivity) [47].

Interesting results were reported by Thorell, who on the basis of studies on persons diagnosed with depression showed that, regardless of the type and severity of the depression disorders, the number of patients with electrodermal hyperactivity—which can be an important predictor of suicidal tendencies, according to this author—doubles versus the control group [31].

In the analysis of the EDA activity profile, age and gender of subjects is of importance. Reports indicate a positive correlation between a general SCL, age, and sex in a group of patients diagnosed with depression. The research shows that women are characterised by a lower general SCL versus men, and the tendency for deterioration in these values increases with age. Possibly, the obtained results are influenced by a decrease in sweat gland activity associated with body ageing [49,50]. The authors in this area emphasise the strong importance of initial SCL values, a feedback loop associated with the speed of emotional changes (activation) and an inhibition ratio negatively proportional to the stimulation level in a general evaluation of electrodermal activity (EDA) measurements. On the basis of the listed parameters, researchers found that reduction in positive emotions is directly proportional to reduced physiological activity [45].

The lateralisation of brain function is also an important parameter in electrodermal activity analysis. Researchers noticed that evaluation of electrodermal activity for verbal tasks is associated with measurements conducted on the right, while for visual tasks these are associated with measurements conducted on the left. The authors showed that people diagnosed with depression are better in the solving of visual tasks versus healthy people, who are better at solving verbal tasks. It is possible that this difference may result from a functional asymmetry of the right and the left sides of the body, which concerns differences in the structure and functions of both cerebral hemispheres (positive–negative asymmetry, automatic-reflexive system of valuation) [51,52].

LeDoux emphasises the significant importance of two of the brain’s emotional memory systems. He associates a role of the hippocampus and cortical areas cooperating with it (emotional aspects) with explicit (declarative) memory, while the implicit (or non-declarative) memory is associated with the amygdala and related areas (body reactions as a response to the emotional aspect). In his opinion, these two systems function simultaneously, their functions are mutually independent, and the registered EDA activity reflects that cooperation [53].

Results of previous studies on electrodermal activity differ, due to incorrect criteria for inclusion into study groups for methodological assumptions. It seems justified that they should consider age, sex, clinical diagnosis, and medicines taken. Including women and men in the same analysis appears to be incorrect, similarly as grouping together patients with first and successive disease episodes [3,20].

From the point-of-view of result reliability, it also appears important to include in the studies patients diagnosed with psychotic and non-psychotic disorders, separate analysis of groups of women and of men, and have patients of a similar age qualify for the research. To determine the EDA profile for a specific group of patients, use of an appropriately selected research procedure is important to consider various deficits in specific clinical states. Only by using such standard procedure can a reliable profile be developed, which in turn will allow for the application of appropriate therapy [20].

The above suggestions are confirmed by Pazderka-Robinson, who emphasised the significant differences in electrodermal activity in different studied groups, similarities in groups with bipolar disorders and depression, and similarities in people with hallucinations, organic disorders, and schizophrenia patients [54].

## 4. Modulation of Electrodermal Activity Based on GSR Biofeedback

GSR Biofeedback using Verim 3.0 is a method of neurorehabilitation used to modulate mental condition. It was proven that certain areas of the brain, mainly substantia nigra (SN) and ventral tegmental area (VTA), play an important role in learning processes, mainly those traditionally conditioned, i.e., paired with a reward. The dopaminergic system, forming a pathway of endogenous regulation, is responsible for their strengthening and coding. The neural activity modulated on the basis of Biofeedback represents an intended control of that regulation, and the method itself, based on the feedback process, significantly improves that ability [1].

There are many forms of Biofeedback and various training devices; one of them is GSR Biofeedback based on the device Verim 3.0 (medical company ELMIKO, Warsaw, Poland). It analyses data obtained during the training concerning the levels of relaxation and activation of the subject. They are processed digitally, and contain the following indicators: a total average, an initial SCL value (total automatic value/start value), a maximum relaxation value (expressed as percentage or as an absolute kOhm value), and a minimum relaxation value, being at the same time a maximum activation value (expressed as percentage or as an absolute kOhm value) [55].

The initial value is created automatically and depends on an activation level in each person at the beginning of a session (dotted line). On this basis, the computer records the mental and physical condition of a patient during each new session. The total average is represented by a continuous line, and this includes the average relative changes in GSR.

Both lines are at different locations and distances than each other, which depend on the initial condition of the subject, and have different max and min values for relaxation and activation. When the dotted line is above the continuous line, this means that the patient has a higher initial SCL value for activation, which decreases during the session. A decrease in the value associated with activation (the continuous line located below the dotted line) means an increase in relaxation, and data forming it concerns average values obtained during the whole session (Figure 3a,b). The decreasing continuous line in the recording, with scarce skin conductance responses (SCRs), indicates that relaxation predominates over activation, while the rising line with the increased number of SCRs proves that activation predominates over relaxation [55,56].

The aim of the GSR-BF session is to stabilise the patient’s mental condition with control and modulation, depending on the mental and physical condition and on deficits notified by the patient. When exercises from different modules are correctly selected, they can be adapted to needs notified by the patient (improvement in cognitive processes, concentration, memory, perceptiveness, executive functions, and stabilisation of emotions) [3,55].

GSR-BF using Verim 3.0 contains several modules. The Energy Room module is associated with energy and contains exercises on relaxation, self-regulation, and concentration. It allows cognitive reaction to feedback from the computer, provided by a physiological parameter. Typical exercises in this module are the “Centre” training, teaching relaxation and self-regulation, and “Balance”, improving concentration [55].

The Resource Room module is associated with resources such as patience or appropriate reaction. A popular exercise in this module is the “Insects” training, in which mental and interactive behaviours are experienced, reflecting at the same time the mental and physiological condition of the patient, similarly to the “Patience” training [55].

The Intelligence Room module was designed to develop self-assessment skills and to strengthen the sense of self-esteem. An interesting training in this module is the “Plot-it” exercise, in which the subject compares the subjectively perceived mental and physical condition with the condition recorded by the computer (plots correlation, self-esteem) [55]. 

The Network Room module represents an interesting solution, facilitating a network group activity (teamwork) together with mental and physical assessment of the training participants. When all modules are considered, the typical programme of exercises is based on the Centre (relaxation), Balance (concentration), and Insects (self-regulation, exercises with positive and negative loops) trainings [55]. 

## 5. Discussion

GSR-BF based on the electrodermal activity (EDA) method indicated that this form of neurorehabilitation is worth noticing, because EDA—as a mental and physiological parameter—reflects basic cognitive processes [38], which can be modified and improved by exercises. This is confirmed by studies demonstrating an overlap of the autonomous SCR amplitude on the central activity associated with the EEG recording (mainly the N200 wave) [57,58].

The importance of synaptic plasticity in these processes is believed to be indicative of functioning of neural circuits and numerous molecular connections [59,60]. This is confirmed by studies using BDNF (brain derived neurotrophic factor), a biomarker whose level indicates a synergism of the central and the peripheral nervous systems responsible for dopaminergic neurotransmission and neurocognitive functions [61,62,63].

GSR Biofeedback, as a non-invasive method, models human brain activity. A graphic recording of electrodermal activity is evidence of this activity and of the synaptic stimulation which—through the long-term potentiation (LTP) process—causes an increase in both the synaptic strength and its biochemical and anatomical changes (synaptogenesis) [55,64,65]. 

The authors in this field compared Biofeedback-based self-regulation to a learning process and instrumental conditioning related to strengthening of specific behaviours and their rewarding. The researchers confirm the effect of the dopaminergic system on encoding of the reward pathway (substantia nigra-SN/tegmental area-VTA) [66,67] and ability to self-regulate SN/VTA based on a positive correlation with skin conductance and emotional stimulation [1]. 

Changes in the electrodermal response were described in detail in the study by Thorell et al. [31]. They conducted an extensive meta-analysis in which they showed the changes in the GSR curve at various stages over the course of affective disorders. Their analysis showed the presence of high hyporeactivity in patients with unipolar and bipolar disorder and confirmed the clearly marked parameters of suicide risk. The author’s research has shown that the greatest changes in low electrodermal reactivity occur in the group of patients with bipolar disorder (80.0%—101 out of 126 patients) than in other affective disorders (66.0%—432 out of 657 patients). The meta-analysis demonstrated by Thorell et al. proved that bipolar affective disorders carry the highest risk of suicide among this group of patients (compared to people diagnosed with schizophrenia—50.0%). Moreover, the authors noted that the transition from hyporesponsivness to reactivity is of particular interest in terms of effective treatment options for psychiatric patients. While there are more publications on EEG-Biofeedback therapy, there is less research related to GSR-Biofeedback therapy, although the main assumption is that training therapy based on analogous feedback reflects the neurophysiological state of patients [68,69]. 

As mentioned, galvanic skin response is a parameter that determines the neurophysiological state of the subjects. This marker has been very useful in recent years, and has been used in many mental [15,25,26,27,30,31,32,33,34,35,36,37,38] and neurological [56,58,70,71] diseases to analyse the stage of the disease or changes occurring under the influence of therapy or treatment. GSR Biofeedback can also be used to describe the effects of Biofeedback therapy by analysing electrodermal activity. 

Based on the results of studies conducted by other authors, mainly Thorella et al., the authors of this study attempted to conduct this type of scientific experiment based on GSR-Biofeedback therapy in patients diagnosed with schizophrenia [31]. The starting point of the analyses was the confirmation of the results of studies by other researchers monitoring electrodermal activity in different periods of the disease. In addition, the authors assumed that, since electrodermal reactivity changes during the disease, the implementation of structured training (relaxing, improving concentration, improving executive functions) accelerates this process, while reducing the symptoms of the disease (in various aspects, mainly cognitive). The results of the research, including the main assumptions, the purpose of the work, the inclusion and exclusion criteria, and the methodology, contain papers that confirm the effectiveness of rehabilitation interventions based on the GSR-Biofeedback method [72,73,74]. 

Summing up the analysis of the reports, it can be said that the GSR-BF profile is a sum of mental and physical responses to external and internal body stimuli, formed by the initial value, SCL activity, skin conductance response (SCRs), and average values. 

As shown before, the low reactivity and the associated reduction in skin conductance responses, SCRs, are characteristic of persons diagnosed with depression, organic and hallucination disorders, and schizophrenia in remission. The high reactivity and the associated numerous skin conductance responses (SCRs) are characteristic of persons with anxiety, patients at a mania stage, and those diagnosed with schizophrenia at the acute stage. Both low and high reactivity indicates disorders in reaction and reduced ability to process information. It is relatively probable that neurorehabilitation based on GSR-BF is a method worth noticing, and that in the future can represent an interesting area of rehabilitation, supplementing a comprehensive treatment of people with mental disorders.

There are many limitations of this work regarding: the type of equipment produced by different companies, laboratory conditions of the laboratories performing training, and the size of the studied groups; nevertheless, the technique of training, graphical record, and criteria for including patients in the study remain the same. Any method that brings benefits to the daily functioning of people who suffer from mental disorders is worth considering [6,7,8,9].

## 6. Conclusions

GSR, as a new form of neurorehabilitation, can complement the primary role of pharmacological treatment. Its wide application, based on psychosomatic feedback, allows for improvement of cognitive and executive functions based on the neuromodulation process. Previous research results confirm its effectiveness in neurological disorders such as stroke, epilepsy, and multiple sclerosis, and somatic disorders such as chronic pain, fibromyalgia, and Raynaud’s disease.

The available literature proves that electrodermal activity is an important parameter in determining the psychophysical state of the respondents, both rehabilitated patients with the GSR Biofeedback method and those monitored during the disease.

## Figures and Tables

**Figure 1 ijerph-19-13428-f001:**
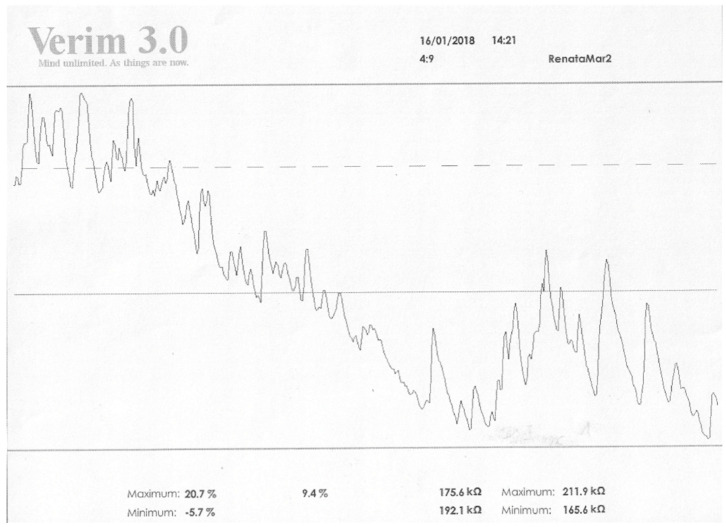
The results obtained in the field of EDA before the GSR Biofeedback therapy. Legend: max vs. min—values of resistance achieved by the patient in kOhm; values on the right refer to relaxation, on the left to activation. Higher values in relaxation training indicate an increase in resistance and thus an increase in relaxation; low values indicate a decrease in resistance, an increase in conductivity, and thus a decrease in relaxation. The values on the left side are the activation level. If relaxation increases, the activation of the organism decreases. Five max values vs. min are the results analogous to those presented in kOhm.

**Figure 2 ijerph-19-13428-f002:**
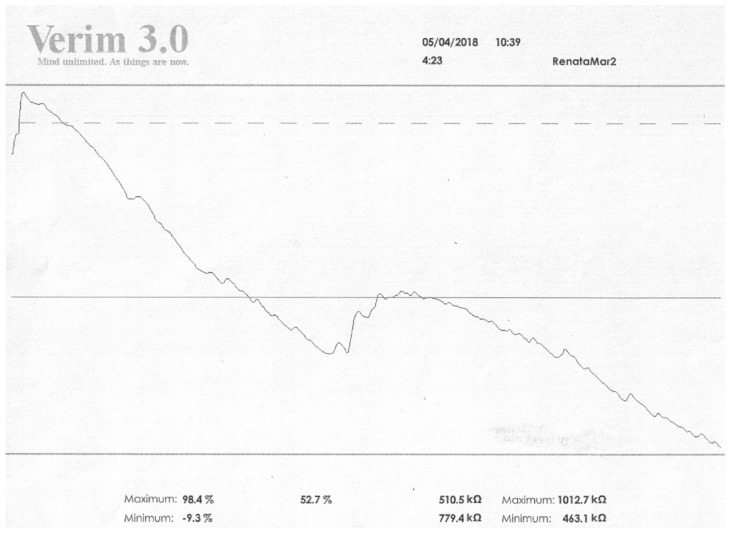
The results obtained in the field of EDA after 3 months of GSR Biofeedback therapy. Legend: max vs. min—values of resistance achieved by the patient in kOhm; values on the right refer to relaxation, on the left to activation. Higher values in relaxation training indicate an increase in resistance and thus an increase in relaxation; low values indicate a decrease in resistance, an increase in conductivity, and thus a decrease in relaxation. The values on the left side are the activation level. If relaxation increases, the activation of the organism decreases. Five max values vs. min are the results analogous to those presented in kOhm.

**Figure 3 ijerph-19-13428-f003:**
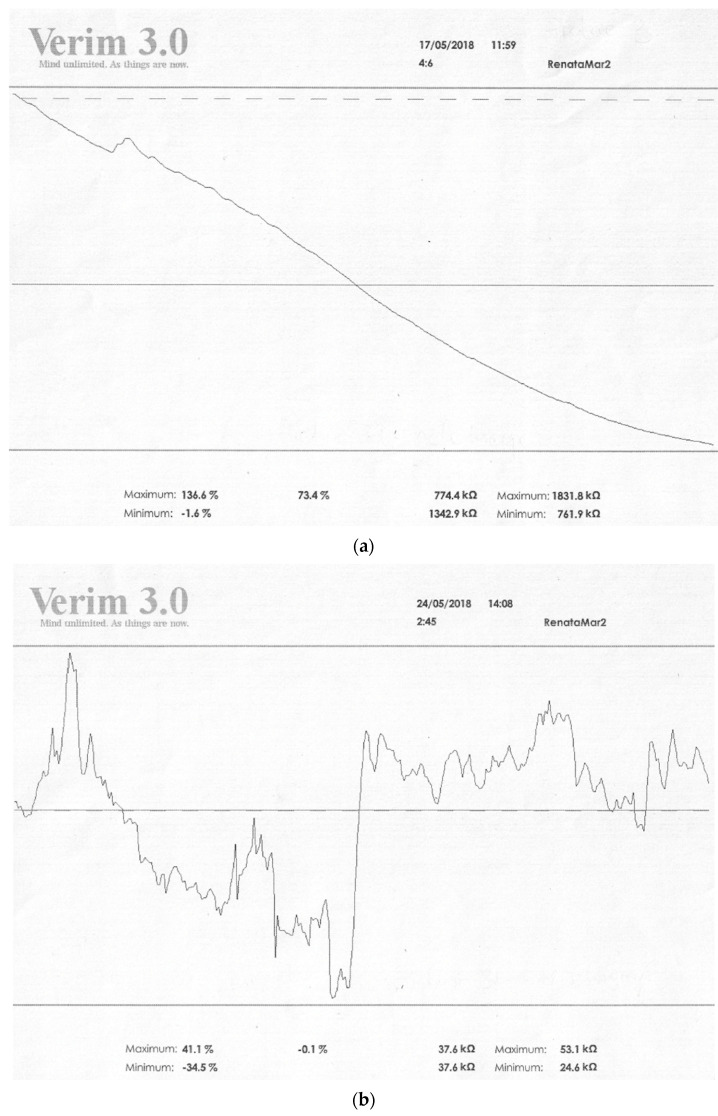
(**a**) Figure showing the state of relaxation. (**b**). Figure showing the state of non-relaxation. Legend: max vs. min—values of resistance achieved by the patient in kOhm; values on the right refer to relaxation, on the left to activation. Higher values in relaxation training indicate an increase in resistance and thus an increase in relaxation; low values indicate a decrease in resistance, an increase in conductivity, and thus a decrease in relaxation. The values on the left side are the activation level. If relaxation increases, the activation of the organism decreases. Five max values vs. min are the results analogous to those presented in kOhm.

**Table 1 ijerph-19-13428-t001:** Mental disorders and the EDA profile (SCL and SCRs).

Type of Mental Disorder	EDA Profile (Skin Conductivity) SCL	EDA Profile (Change in Conductivity) SCR	Ref.
Anxiety disorders	Reduced	SCR lability	[15,25,26]
Depressive disorders	Reduced, “flat profile”, deficit of psychological flexibility	Lowered or elevated (remission period)	[27,28,29,30]
Suicidal tendencies	Lowered	Lowered	[31,32]
Bipolar affective disorder	Value fluctuations, decreased in depression, increased in mania	Stabilization of values in the period of remission	[33,34,35]
Schizophrenia	Labile or lowered values (spontaneous, non-specific reaction)	Lowered values (dominance of negative symptoms) or increased values (dominance of positive symptoms)	[36,37,38,39]

Low or high values indicate a disturbed reaction and a reduced ability to process information.

## Data Availability

Not applicable.

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
