# Peer review of "Galvanic Skin Response Features in Psychiatry and Mental Disorders: A Narrative Review"

_ijerph, 2022, doi:10.3390/ijerph192013428_

Round 1
Reviewer 1 Report (Previous Reviewer 1)
the improved form is satisfactory and I think it is suitable for publication.
Author Response
Dear reviewer,
Thank you very much for your kindness and for the work put into making the review.
Kind regards,
Agnieszka Markiewicz-Gospodarek
Reviewer 2 Report (New Reviewer)
At this stage, this paper is providing a shadow information about the Galvanic Skin Responses in Psychiatry.
Psychiatry issues are very complex and very interfering with other medical conditions and environmental factors. Therefore, a sensation by biofeedback will give only superficial descriptions to non-benefits in practical way.
Therefore, to be fair, I will not reject this paper at the moment. But I encourage you that you may convert your narrative review to scoping review or to systematic review. It is much better if you do that. Because I need to see more comparative tables among studies and I encourage you to assess each study you have selected in regards bias, reliability, and evidences. Also, I recommend you to apply JBI critical appraisal to each reference.
Author Response
Dear Reviewer,
The authors of the publication titled: "Galvanic Skin Response Features in Psychiatry and Mental Disorders: A Narrative Review" would like to thank you very much for the time spent by the reviewers in processing the article. In accordance with the reviewers' suggestions, the authors of the paper have made every effort to make corrections, in accordance with the original version of the review. It was not the intention of the paper's authors to perform a meta-analysis, which is again requested by one of the reviewers. The purpose of the paper was to present a new, innovative method of rehabilitation based on electrodermal activity, a detailed description of it, and the principles of its interpretation in various psychiatric disorders. Therefore, to continue to make corrections requested by only one of the reviewers is tantamount to the authors withdrawing the article for further proceedings, since a complete revision of the article, i.e., a meta-analysis from the beginning, was not the authors' goal and will result in a change in the concept of the article.
Kind regards
Agnieszka Markiewicz-Gospodarek
This manuscript is a resubmission of an earlier submission. The following is a list of the peer review reports and author responses from that submission.
Round 1
Reviewer 1 Report
A very interesting narrative review about GSR Biofeedback in Psychiatry . The authors properly approach the subject with detail and analysis. Biofeedback as modern methods of neurorehabilitation method in persons with mental disorders as method of providing a person with information regarding the state of their body using various devices in the form of light, sound, or other signals.
Abstract is informative. The method is suitable to investigate the research question and test the hypotheses. Introduction provide a concise background. The authors explain and interpret the results satisfactorily References are well written.
Author Response
Agnieszka Markiewicz – Gospodarek
Medical University of Lublin
Department of Human Anatomy
Jaczewskiego 4
20-090 Lublin, Poland
agnieszkamarkiewiczgospodarek@umlub.pl
Dear Reviewer,
Thank you very much for reviewing our manuscript. We appreciate the interest and commitment you have provided for this work. We are very grateful for your extremely precious comments. We are convinced that thanks to your suggestions this manuscript will be much more valuable.
We are pleased to submit explanations and details of our revisions in the manuscript entitled “Galvanic Skin Response (GSR) Biofeedback in Psychiatry”.
We hope that after this revision, the manuscript is of a higher quality and worth reading.
We wish you all the best!
Sincerely,
Agnieszka Markiewicz – Gospodarek
on behalf of all authors
Reviewer 2 Report
The authors clearly expound on the topic . It would be appropriate to initially describe how the search for articles to be included in review was conducted. Perhaps it would be helpful to write Galvanic Skin Response (GSR) in the title and not just GSR.
Author Response
Agnieszka Markiewicz – Gospodarek
Medical University of Lublin
Department of Human Anatomy
Jaczewskiego 4
20-090 Lublin, Poland
agnieszkamarkiewiczgospodarek@umlub.pl
Dear Reviewer,
Thank you very much for reviewing our manuscript. We appreciate the interest and commitment you have provided for this work. We are very grateful for your extremely precious comments. We are convinced that thanks to your suggestions this manuscript will be much more valuable.
We are pleased to submit explanations and details of our revisions in the manuscript entitled “Galvanic Skin Response (GSR) Biofeedback in Psychiatry”.
The followings are our point-by-point responses:
- We developed the abbreviation in the title as suggested.
We hope that after this revision, the manuscript is of a higher quality and worth reading.
We wish you all the best!
Sincerely,
Agnieszka Markiewicz – Gospodarek
on behalf of all authors
Reviewer 3 Report
I understand that biofeedback therapy using GSR as a new treatment modality is attracting a lot of attention and is very promising at a time when effective treatments for intractable mental disorders are limited. It is therefore of importance to summarise GSR biofeedback therapy for psychiatric disorders.
The authors provide a clear summary of the characteristics of GSRs in different psychiatric disorders.
As the title was "GSR Biofeedback in Psychiatry", I was interested to see what GSR biofeedback practices are actually in place for each psychiatric disorder, but the description of this part seemed limited.
Author Response
Agnieszka Markiewicz – Gospodarek
Medical University of Lublin
Department of Human Anatomy
Jaczewskiego 4
20-090 Lublin, Poland
agnieszkamarkiewiczgospodarek@umlub.pl
Dear Reviewer,
Thank you very much for reviewing our manuscript. We appreciate the interest and commitment you have provided for this work. We are very grateful for your extremely precious comments. We are convinced that thanks to your suggestions this manuscript will be much more valuable.
We are pleased to send a revised version of the manuscript entitled “Galvanic Skin Response (GSR) Biofeedback in Psychiatry”.
The followings are our point-by-point responses:
- We presented the application of GSR Biofeedback therapy in various mental disorders in the form of a table.
We hope that after this revision, the manuscript is of a higher quality and worth reading.
We wish you all the best!
Sincerely,
Agnieszka Markiewicz – Gospodarek
on behalf of all authors
Round 2
Reviewer 3 Report
I appreciate your revised work. The table would let readers understand how different each mental disorder has the GSR features.
I don't think this manuscript sufficiently includes GSR "biofeedback therapy" for each mental disorder. I found that they were mainly GSR features in mental disorders. Even if the manuscripts included GSR biofeedback therapy, some vital information might be missing such as how the GSR value was feedbacked to a participant, the difference in clinical rating scale before and after biofeedback therapy, etc.
Author Response
Dear Reviewer,
Thank you very much for reviewing our manuscript. We appreciate the interest and commitment you have provided for this work. We are very grateful for your extremely precious comments. We are convinced that thanks to your suggestions this manuscript will be much more valuable.
We are pleased to submit explanations and details of our revisions in the manuscript entitled “Galvanic Skin Response features in psychiatry and mental disorders: A Narrative Review”.
The followings are our point-by-point responses:
1. Electrodermal activity is a parameter that is used to assess a patient’s psychophysiological state. It has applications in various clinical entities, at different stages of disease and during the application of various forms of therapeutic interventions (e.g., Biofeedback). The authors of the publication unanimously agreed that the reviewer’s 3 comment is justified, since a significant part of the work concerns the description of principles of interpretation of the GSR curve i.e., its features. Therefore, after making the necessary corrections, the authors of the paper accepted the proposal to change the title of the paper, as suggested by the reviewer.
We hope that after this revision, the manuscript is of a higher quality and worth reading.
We wish you all the best!
Sincerely,
Agnieszka Markiewicz – Gospodarek
on behalf of all authors